



# Assessing the role and consistency of satellite observation products in global physical-biogeochemical ocean reanalysis

David Ford[1]

[1]Met Office, FitzRoy Road, Exeter, EX1 3PB, UK

**Correspondence:** David Ford (david.ford@metoffice.gov.uk)

**Abstract.** As part of the European Space Agency's Climate Change Initiative, new sets of satellite observation products have been produced for Essential Climate Variables including ocean colour, sea surface temperature, sea level and sea ice. These new products have been assimilated into a global physical-biogeochemical ocean model, to create a set of 13-year reanalyses at 1° resolution and 3-year reanalyses at 1/4° resolution. In a series of experiments, the variables were assimilated individually and in combination, in order to assess their consistency from a data assimilation perspective. The satellite products, and the reanalyses assimilating them, were found to be consistent in their representation of spatial features such as fronts, sea ice extent and bloom activity. Assimilating multiple variables together often resulted in larger mean increments for a variable than assimilating it individually, revealing ways in which the model and assimilation scheme could be improved. Sea surface fugacity of carbon dioxide had lower errors against independent observations in the higher resolution simulations, and was improved by assimilating ocean colour or sea ice concentration, but degraded by assimilating sea surface temperature or sea level anomaly. Phytoplankton biomass correlated more strongly with net air-sea heat fluxes in the reanalyses than chlorophyll concentration did, and the correlation was weakened by assimilating ocean colour data, suggesting that studies of phytoplankton bloom initiation based solely on chlorophyll data may not provide a full understanding of the underlying processes.

## 1 Introduction

In order to understand and monitor the Earth's climate system, it must be routinely and systematically observed. Satellite remote sensing is particularly valuable in providing daily global data, with satellite records for some variables dating back to the 1970s. To allow the detection of climate variability and change, long-term continuous time series of such data need to be consistently processed, stable, and free from artificial trends. Products which match these requirements can be referred to as climate data records (CDRs) (NRC, 2004). To address this requirement for a set of essential climate variables (ECVs) (GCOS, 2011) that can be observed from space, the European Space Agency (ESA) initiated a programme called the Climate Change Initiative (CCI) (Plummer et al., 2017). For ECVs including ocean colour, sea surface temperature, sea level and sea ice, sets of satellite observation products have been developed and made available, with the aim that they can be used as CDRs. A further aim is for the CCI products to be consistent between ECVs, allowing an integrated assessment of the climate system.

Such observation products are vital for understanding climate variability and change, but are insufficient on their own. Coverage is incomplete, not all variables of interest are routinely observed, and there is no predictive capability. Models are



required to address these aspects, and in conjunction with observations can provide a much fuller understanding of the Earth system. Recognising this need, the CCI programme includes a Climate Modelling User Group (CMUG), to assess CCI data products from a modelling perspective.

A powerful way to combine these sources of information is through data assimilation, which can be used to create reanalyses (Storto et al., 2019). As with CDRs, reanalyses are of most benefit to climate studies if they are stable and consistent, both throughout time and between model variables. In these cases, reanalyses can be used to provide valuable insights into the Earth system, which may not be available from observations or models alone (Jackson et al., 2016). However, models have limitations, and even with the aid of data assimilation cannot accurately simulate all aspects of the climate system. Nevertheless, the process of creating and assessing reanalyses itself can aid understanding, both of the real world and of the underlying models. This can lead to improvements in these models, in turn leading to improvements in the next generation of reanalyses and climate projections.

Physical ocean reanalyses are increasingly used for climate studies (Balmaseda et al., 2015; Storto et al., 2019), and physical-biogeochemical reanalyses assimilating ocean colour data on its own are starting to be developed (Ciavatta et al., 2016; Ford and Barciela, 2017; Fennel et al., 2019). It is not yet routine though to combine the assimilation of physics and biogeochemistry in a single ocean reanalysis. A major reason for this is that, especially in global models, assimilating physical observations has been widely found to degrade biogeochemical fields (While et al., 2010; Raghukumar et al., 2015; Park et al., 2018) through the creation of spurious vertical mixing. This is an outstanding issue for the ocean data assimilation community, and has not been solved in this study. However, whilst this currently limits the ability to produce stable long-term physical-biogeochemical ocean reanalyses, consistency of variability and features can still be explored, especially if the biogeochemistry is constrained by assimilation of ocean colour data.

This paper presents a set of physical-biogeochemical ocean model runs, assimilating different combinations of satellite data products, as well as in situ temperature and salinity observations. These were performed as part of the CCI CMUG. The aims of the study were:

- To assess whether the four marine ECVs developed as part of phase one of CCI are mutually consistent from a data assimilation perspective.

- To assess the consistency of physical-biogeochemical relationships in reanalyses assimilating different combinations of these ECVs.

- To assess the impact of assimilating the ECVs on the marine carbon cycle.

To address these aims, the model runs have been assessed in a variety of ways. Results are presented here evaluating the assimilation increments, frontal features in the Agulhas Current region, sea ice extent and phytoplankton blooms in the Arctic, validation against in situ carbon observations, and the relationship between phytoplankton variability and air-sea heat fluxes.



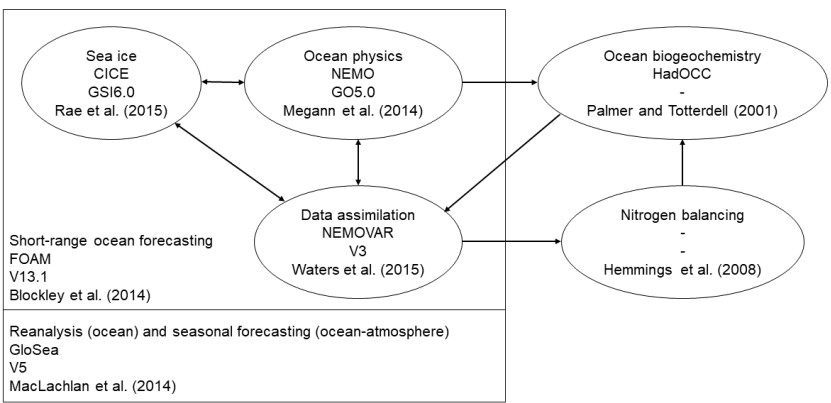

**Figure 1.** Model and assimilation setup used. Arrows denote direction of coupling.

## 2   Model and assimilation

The model and assimilation setup used in this study, and the couplings between different components, is shown schematically in Fig. 1. An overview of the components is given below, for further details the reader is referred to the references provided.

The physical modelling system was a reanalysis version of v13.1 of the global Forecasting Ocean and Assimilation Model (FOAM), which is run operationally at the Met Office (Blockley et al., 2014). FOAM v13.1 includes the GO5.0 configuration (Megann et al., 2014) of the Nucleus for European Modelling of the Ocean (NEMO) (Madec, 2008) hydrodynamic model, the GSI6.0 configuration (Rae et al., 2015) of the Los Alamos sea ice model (CICE) (Hunke and Lipscomb, 2010), and a 3D-Var configuration of the NEMOVAR v3 data assimilation scheme (Waters et al., 2015). FOAM v13.1 is a development of
the FOAM v12.0 system described by Blockley et al. (2014), and the differences between the two are detailed by Jackson et al. (2016). It is the same configuration that was used to produce the GloSea5 physical ocean reanalysis (MacLachlan et al., 2015; Jackson et al., 2016), with atmospheric forcing provided by the ERA-Interim reanalysis (Dee et al., 2011).

The biogeochemical model used was the Hadley Centre Ocean Carbon Cycle model (HadOCC) (Palmer and Totterdell, 2001). HadOCC is a relatively simple nutrient-phytoplankton-zooplankton-detritus (NPZD) model, with a fully coupled carbon
cycle and variable carbon-to-chlorophyll ratio. FOAM-HadOCC has been used for previous assimilation studies (Ford et al., 2012; While et al., 2012; Ford and Barciela, 2017). The HadOCC model settings used in this study were the same as used by Ford and Barciela (2017), except that atmospheric $CO_2$ concentrations were taken from the National Oceanic and Atmospheric Administration Earth System Research Laboratory (NOAA/ESRL) global monthly mean observation product (Dlugokencky and Tans, https://www.esrl.noaa.gov/gmd/ccgg/trends).

Assimilation of physics data followed the method described by Waters et al. (2015), with the added option to use multiple correlation length scales developed by Mirouze et al. (2016). The assimilation of ocean colour data was the same as the



approach taken by Ford et al. (2012) and Ford and Barciela (2017), except that the analysis correction scheme of Martin et al. (2007) was replaced by the 3D-Var NEMOVAR implementation of Waters et al. (2015). NEMOVAR was used to create an analysis of surface $\log_{10}$(chlorophyll), and the nitrogen balancing scheme of Hemmings et al. (2008) was used to create a set

of 3D increments to all the biogeochemical state variables, which were applied to the model. Specific details regarding the assimilation of each ECV are given in the following sections.

## 3    Observations

This study used various sets of satellite and in situ observation products, for both assimilation and validation, and these are described in turn below. Product versions used were the most recent official releases available at the time the study began.

Satellite ocean colour (OC) data were taken from the v2 CCI product (Sathyendranath et al., 2017). This study used level three (L3) daily average chlorophyll data, merging information from the Sea-viewing Wide Field-of-view Sensor (SeaWiFS) on board SeaStar, the Moderate Resolution Imaging Spectroradiometer (MODIS) on board Aqua, and the Medium Resolution Imaging Spectrometer (MERIS) on board Envisat. During the product generation MERIS and MODIS remote sensing reflectances were bias corrected against SeaWiFS, in order to minimise inter-sensor differences. The product also includes

per-pixel uncertainty estimates, following the approach of Moore et al. (2009).

     Satellite sea surface temperature (SST) data were taken from the v1.1 CCI product (Merchant et al., 2014). This study used level two pre-processed (L2P) SST data from the Advanced Very High Resolution Radiometer (AVHRR) series of sensors, and level three uncollated (L3U) SST data from the Along-Track Scanning Radiometer (ATSR) series of sensors. Individual sensors were processed and provided separately, with AVHRR bias corrected against ATSR. Per-pixel uncertainty estimates

are included, as described by Bulgin et al. (2016a, b).

     Satellite sea level anomaly (SLA) data were taken from the v1.1 CCI product (Ablain et al., 2015; https://doi.org/10.5270/esa-sea_level_cci-1993_2014-v_1.1-201512). This study used the Fundamental Climate Data Record (FCDR) product, which provides level 2 (L2) single sensor along-track SLA data for all available sensors. SLA was calculated using the DTU10 (Andersen and Knudsen, 2009; Andersen, 2010) mean sea surface as a reference.

Satellite sea ice concentration (SIC) data were taken from the v1.2 reprocessed global level four (L4) product (Tonboe et al., 2016) of the EUMETSAT Ocean and Sea Ice Satellite Application Facility (OSI SAF), based on the Special Sensor Microwave Imager/Sounder (SSMIS), the Special Sensor Microwave/Imager (SSM/I), and the Scanning Multichannel Microwave Radiometer (SMMR) sensors. An OSI SAF product was used rather than CCI because the CCI SIC project began later than the other ECVs, and an appropriate CCI product for these experiments was not yet available when the study began. However, the

OSI SAF and CCI SIC projects are led by the same group, with shared research and development capabilities. Furthermore, OSI SAF products provide a consistent time series from 1979, whereas CCI products use sensors which only provide a consistent time series from 2002, meaning the OSI SAF products are more appropriate for many reanalysis users.





In situ SST data were taken from v2.5 of the International Comprehensive Ocean–Atmosphere Data Set (ICOADS) (Woodruff et al., 2011) for the period 1998–2007, and from the Global Telecommunications System (GTS) from 2008 onwards. They in-

clude measurements from moored buoys, drifting buoys, and ships (Blockley et al., 2014; Jackson et al., 2016).

In situ temperature and salinity profile (T&S) data were taken from the EN4.1.1 product (Good et al., 2013) with Gouretski and Reseghetti (2010) corrections.

In situ surface fugacity of carbon dioxide ($fCO_2$) data were taken from the v2 Surface Ocean $CO_2$ Atlas (SOCAT) product (Bakker et al., 2014). The capability to assimilate these data exists (While et al., 2012), but in this study $fCO_2$ observations

were only used for validation.

## 4  Experiments

The aims of this study included assessing both spatial features and interannual variability. The former requires a high enough model resolution to represent such features, and the latter requires multi-year model runs. Due to the computational expense of the assimilative coupled physical-biogeochemical model, multiple long runs were unable to be performed at eddy-permitting

resolution. Therefore, two sets of model runs were performed at different resolutions. To assess interannual variability, a set of 13-year runs were performed at 1° horizontal resolution, covering the period 1 January 1998 to 31 December 2010, which are the full years for which there is data from all four satellite ECV products used in this study. To assess spatial features, a set of 3-year runs were performed at 1/4° horizontal resolution, covering the period 1 January 2008 to 31 December 2010. The same vertical resolution was used in both cases, with 75 levels and a 1 m surface layer.

The same CICE and HadOCC settings were used for both resolutions, but some NEMO settings needed to be changed for running at 1° resolution, compared with the 1/4° configuration described by Megann et al. (2014). For the 1° runs, the eddy induced velocity parameterisation of Gent and Mcwilliams (1990) was turned on, and laplacian rather than bilaplacian lateral iso-level diffusion was used on momentum, with associated mixing coefficients varying in 3D rather than 2D. Furthermore, the special treatment of tidal mixing in the Indonesian Throughflow, as developed by Koch-Larrouy et al. (2008), was only used at

1/4° resolution.

Prior to assimilation, observations of physical variables were quality controlled in the same manner as for the operational FOAM system and GloSea5 reanalysis (Blockley et al., 2014; Storkey et al., 2010). OC observations were quality controlled as described for CCI products by Ford and Barciela (2017). All assimilated observations were median-averaged to a resolution of 13 km for the 1/4° runs, and 100 km for the 1° runs. For physics observations, the error variances used by NEMOVAR were

those described by Blockley et al. (2014) for the 1/4° runs, interpolated to 1° for the 1° runs. For OC observations, the error variances were those described for CCI products by Ford and Barciela (2017) for the 1° runs, interpolated to 1/4° for the 1/4° runs. For the 1/4° runs, two correlation length scales were used by NEMOVAR for physical ocean variables, as described by Mirouze et al. (2016). For OC and SIC at 1/4° resolution, and all variables at 1° resolution, a single correlation length scale was used, based on the first baroclinic Rossby radius, as described by Waters et al. (2015). Unlike the operational FOAM

system and GloSea5 reanalysis, no bias correction of SST observations was performed, in order to assess whether the bias





**Table 1.** Summary of model runs performed. "SST" refers to satellite data only; in situ SST and profile data are included with "T&S". HIGH and LOW are intended as relative terms.

| Run identifiers (HIGH = 1/4°, LOW = 1°) | Assimilating | | | | |
|---|---|---|---|---|---|
| | OC | SST | SIC | SLA | T&S |
| HIGH_FREE LOW_FREE | | | | | |
| HIGH_OC LOW_OC | X | | | | |
| HIGH_SST LOW_SST | | X | | | |
| HIGH_SIC LOW_SIC | | | X | | |
| HIGH_SLA LOW_SLA | | | | X | |
| HIGH_OC_SST_SIC LOW_OC_SST_SIC | X | X | X | | |
| HIGH_OC_SST_SIC_SLA LOW_OC_SST_SIC_SLA | X | X | X | X | |
| HIGH_OC_SST_SIC_SLA_T&S LOW_OC_SST_SIC_SLA_T&S | X | X | X | X | X |

correction and processing performed during the creation of the SST CDR was sufficient to allow a stable reanalysis. However, SLA bias correction (Lea et al., 2008; Blockley et al., 2014) was still performed, using the DTU10 (Andersen and Knudsen, 2009; Andersen, 2010) mean dynamic topography (MDT), as this corrects for errors in the MDT.

For the 1/4° runs, spun-up physical and biogeochemical model fields were taken from a previous project, and used as initial

conditions with no further spin-up. For the 1° runs, a dedicated spin-up was performed without data assimilation, covering the period 1 January 1980 to 31 December 1997. The initial conditions were the same as for the spin-up of Ford and Barciela (2017), except for temperature and salinity, which in this study were taken from the EN4.1.1 objective analysis for January 1980 (Good et al., 2013; Gouretski and Reseghetti, 2010), and dissolved inorganic carbon (DIC) and alkalinity, which were taken from the Global Data Analysis Project (GLODAP) climatology (Key et al., 2004), and converted from per unit mass to

per unit volume, to match the units used by HadOCC. At the end of the spin-up, the NEMO sea surface height (SSH) fields were uniformly adjusted to have zero global mean, as the global mean SSH had drifted and would have caused a large initialisation shock when SLA assimilation began. These fields were then used as initial conditions for the main 1° runs.

The model runs performed are summarised in Table 1. At each resolution there were eight runs: a free run with no data assimilation, four runs assimilating each satellite ECV individually, a run assimilating all four satellite ECVs together, a run assimi-



lating all four satellite ECVs plus in situ SST and temperature and salinity profiles (T&S), and a run assimilating satellite SST, OC, and SIC. Unique identifiers for each run are detailed in Table 1, prefixed with "HIGH" for 1/4° and "LOW" for 1°, which are simply intended as relative terms. The identifiers include the variables assimilated (e.g., HIGH_OC_SST_SIC_SLA_T&S for the 1/4° run assimilating all data), with in situ SST and profile data included with T&S.

In most cases, assimilation increments were applied at all model grid points. However, for model stability a few exceptions

were required. No increments were applied in the Baltic Sea in the 1° runs, which is treated as an enclosed sea at this resolution. No assimilation was performed on 18 January 2000 in 1° runs including SLA assimilation, as a few large SLA observations were causing the model to fail. On a few occasions the assimilation caused LOW_SLA and LOW_OC_SST_SIC_SLA_T&S to fail near the Antarctic coast; in these cases no increments were applied for a short period in the surrounding region. Similarly, no increments were applied in the Malvinas Current region on a small number of dates in HIGH_SLA and HIGH_OC_SST_SIC_SLA_T&S.

On all dates, no biogeochemical increments were applied in grid boxes with SIC greater than 0.01, which is a relaxation of the conditions imposed by Ford et al. (2012) and Ford and Barciela (2017). Furthermore, phytoplankton nitrogen increments were limited in magnitude to 1.0 mmol m$^{-3}$ in a region surrounding the Amazon river outflow, prior to running the Hemmings et al. (2008) nitrogen balancing scheme, in order to avoid spuriously large DIC increments at very low chlorophyll concentrations. These cases were generally indicative of issues with the model and assimilation procedure under specific circumstances, rather

than of errors in the observation products.

## 5   Results

The model runs have been assessed through a series of case studies, presented in turn below. These are intended to explore physical-biogeochemical relationships in the model and observations, and the impact of data assimilation on these, rather than simply validating the accuracy of the reanalyses. For validation of the underlying system, the reader is referred to Blockley

et al. (2014) for the physical model and assimilation, Ford and Barciela (2017) for the biogeochemical model and assimilation, and Lea et al. (2014) for data withholding experiments performed with the physics-only system.

### 5.1   Assimilation increments (Fig. 2 and 3)

A measure of how hard the data assimilation is working is the long-term mean and standard deviation of the assimilation increments. This can also provide important information about model biases. The larger the increments, the larger the corrections

being applied to the model to keep it close to the observations. In theory, if the observation products are providing consistent information, and the model and assimilation scheme are performing as intended, then assimilating multiple ECVs should result in smaller mean increments for a given ECV compared with assimilating that ECV alone. To test that theory, maps of the mean increments for 1998–2010 are plotted for OC (Fig. 2) and SST (Fig. 3), from each of the 1° runs assimilating those ECVs. For runs assimilating multiple ECVs, the difference in the absolute mean increments from the run assimilating the single ECV is

also plotted. Note that there is no feedback from the biogeochemistry to the physics, so assimilating OC has no impact on the


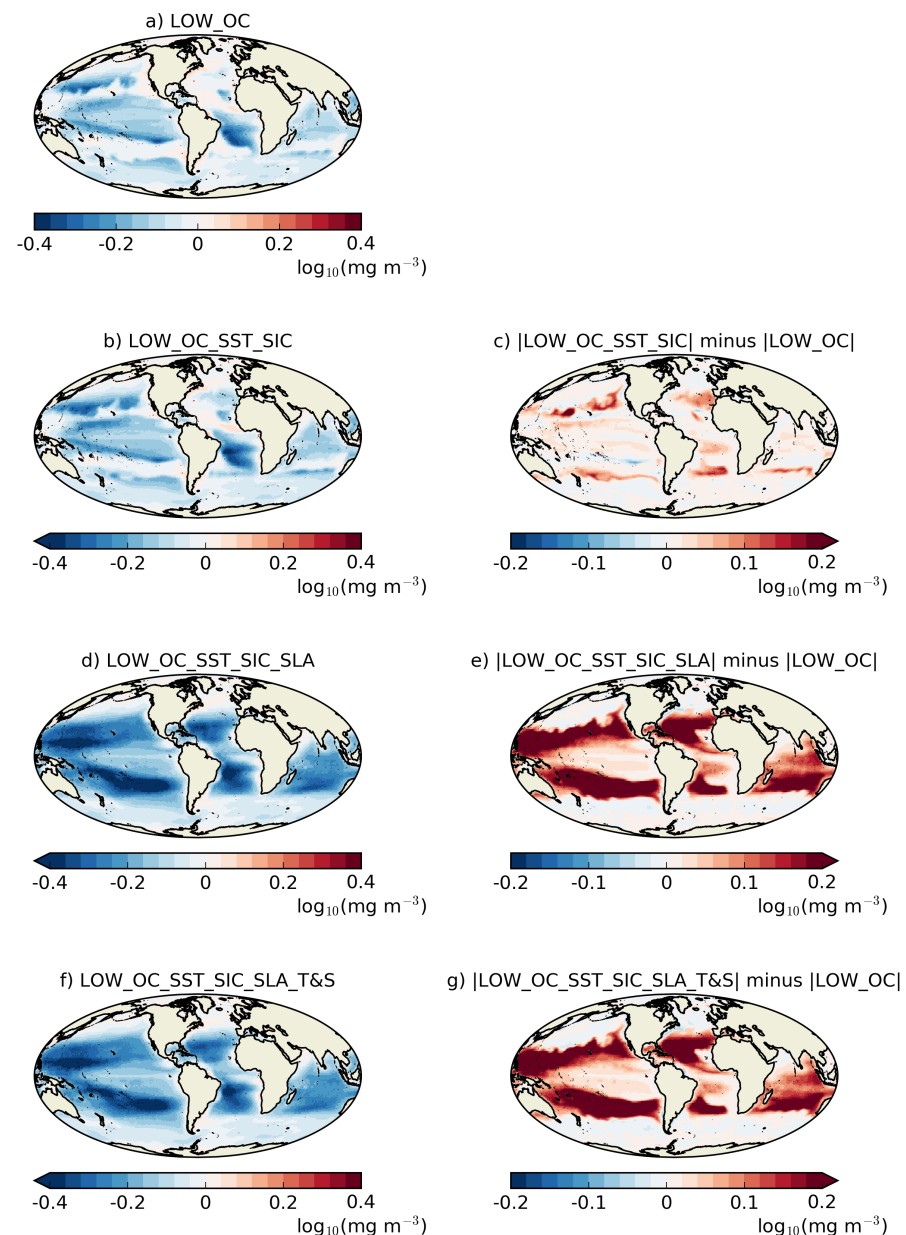

**Figure 2.** Mean surface $\log_{10}$(chlorophyll) assimilation increments for 1998–2010.

physics variables. Mean increments from the 1/4° runs (not shown) showed generally consistent patterns with those from the 1° runs, apart from when assimilating SLA, as discussed below.

When OC was assimilated individually, the mean surface $\log_{10}$(chlorophyll) increments were negative across most of the ocean (Fig. 2a), indicating a persistent model bias which the assimilation was continually trying to correct. This is consistent





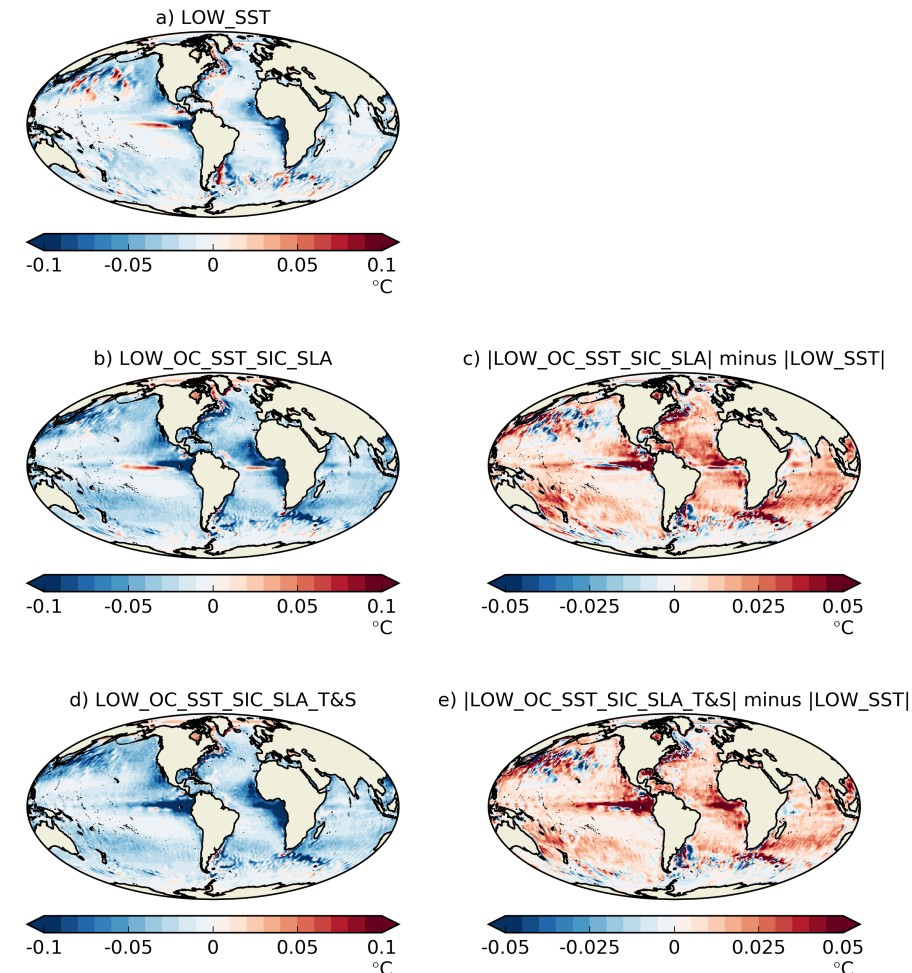

**Figure 3.** Mean SST assimilation increments for 1998–2010.

with the validation presented in Ford and Barciela (2017). When SST was also assimilated (Fig. 2b-c), the magnitude of the mean increments was slightly increased, particularly around the edges of the subtropical gyres. In LOW_FREE the global mean SSH drifted over time due to a freshwater imbalance in the model (Blockley et al., 2014). Assimilating SST without also assimilating T&S is known to generate spurious heat beneath the mixed layer, which served to reduce the drift in SSH, but by introducing a compensating error. This resulted in the spatial extent of the oligotrophic gyres spuriously shrinking, leading to excess primary production which the OC assimilation was continually trying to reduce. Globally, the magnitude of the $\log_{10}$(chlorophyll) increments was then increased considerably when SLA was also assimilated (Fig. 2d-e), and remained increased when T&S was assimilated (Fig. 2f-g). The reason for this is the issue mentioned in the introduction, and explored by While et al. (2010) and Park et al. (2018), that assimilating SLA and T&S results in spurious vertical mixing, especially in equatorial regions, bringing excess nutrients to the surface and fuelling primary production. Counterintuitively, despite the



biggest impact on mixing being at the equator, the biggest impact on the $\log_{10}$(chlorophyll) increments was away from the equator. This is because phytoplankton growth in these runs was not generally nutrient-limited around the equator, so an increased nutrient supply had little impact on chlorophyll concentration. However, over the 13 years of the runs these nutrients were advected away from the equator into surrounding nutrient-limited regions, resulting in excessive primary production and an increase in the mean increments as the assimilation tried to correct this.

When SST was assimilated individually, the mean SST increments were also generally negative (Fig. 3a), indicating a persistent warm bias in the model, consistent with the findings of Blockley et al. (2014). When SLA assimilation was also introduced (Fig. 3b-c), the mean increments increased in magnitude, apparently due to the subsurface density corrections applied by the SLA assimilation not being entirely consistent with the SST, meaning larger SST increments were required to correct the bias. It has previously been shown by Lea et al. (2014) that SLA assimilation is best performed in combination with

T&S assimilation, as this allows subsurface density errors in the model to be corrected. Adding T&S assimilation (Fig. 3d-e) reduced the mean increments, but they remained larger than when assimilating SST alone.

When SLA was assimilated individually (not shown), the SLA increments were largest in energetic regions such as around the equator and western boundary currents, as well as the Southern Ocean. There was no clear global bias, due to the SLA bias correction scheme (Lea et al., 2008). In the 1° model, adding SST assimilation had little impact on the increments, whilst

adding T&S assimilation resulted in patchy increases in mean SLA increments. In the 1/4° model, adding SST assimilation served to reduce the mean SLA increments, and adding T&S assimilation greatly reduced them further. This latter result is consistent with Lea et al. (2014), and highlights the complementarity of SLA and T&S data. The contrasting findings in the 1° model are likely due to the coarse resolution, which is unable to resolve eddies, and the fact that the error covariances were not tuned for the lower resolution.

Assimilating SIC and SST in combination reduced both the mean SIC and SST increments applied in the Antarctic, compared with assimilating SIC and SST individually, but not in the Arctic, where the impact was minimal (not shown).

Given these issues, looking at mean increments does not provide evidence either way about whether the CCI products are mutually consistent, but it does highlight issues with the multivariate assimilation which can be addressed during future development work. It should also be noted that the physics data assimilation is designed to work best when all data types are

available, as these provide complementary information (Lea et al., 2014).

## 5.2 Fronts and eddies (Fig. 4)

An important way in which CDRs and reanalyses should be consistent is in the representation of spatial features, such as the positioning of fronts and eddies, and details of major currents such as the Agulhas. This can be assessed by analysing the horizontal gradients of different fields. Following the method of Martin (2016), horizontal gradients of SST, SLA and surface

$\log_{10}$(chlorophyll) were calculated by bilinearly interpolating model fields to observation locations, binning the resulting model and observation values onto a regular grid over a month, and calculating their spatial derivative. Gradients have been plotted for the final month, December 2010, in the Agulhas Current region, for the CCI products and the 1/4° free run and runs assimilating SST, SLA and OC individually and in combination (Fig. 4). SST and surface $\log_{10}$(chlorophyll) were binned onto a 1/4° grid,





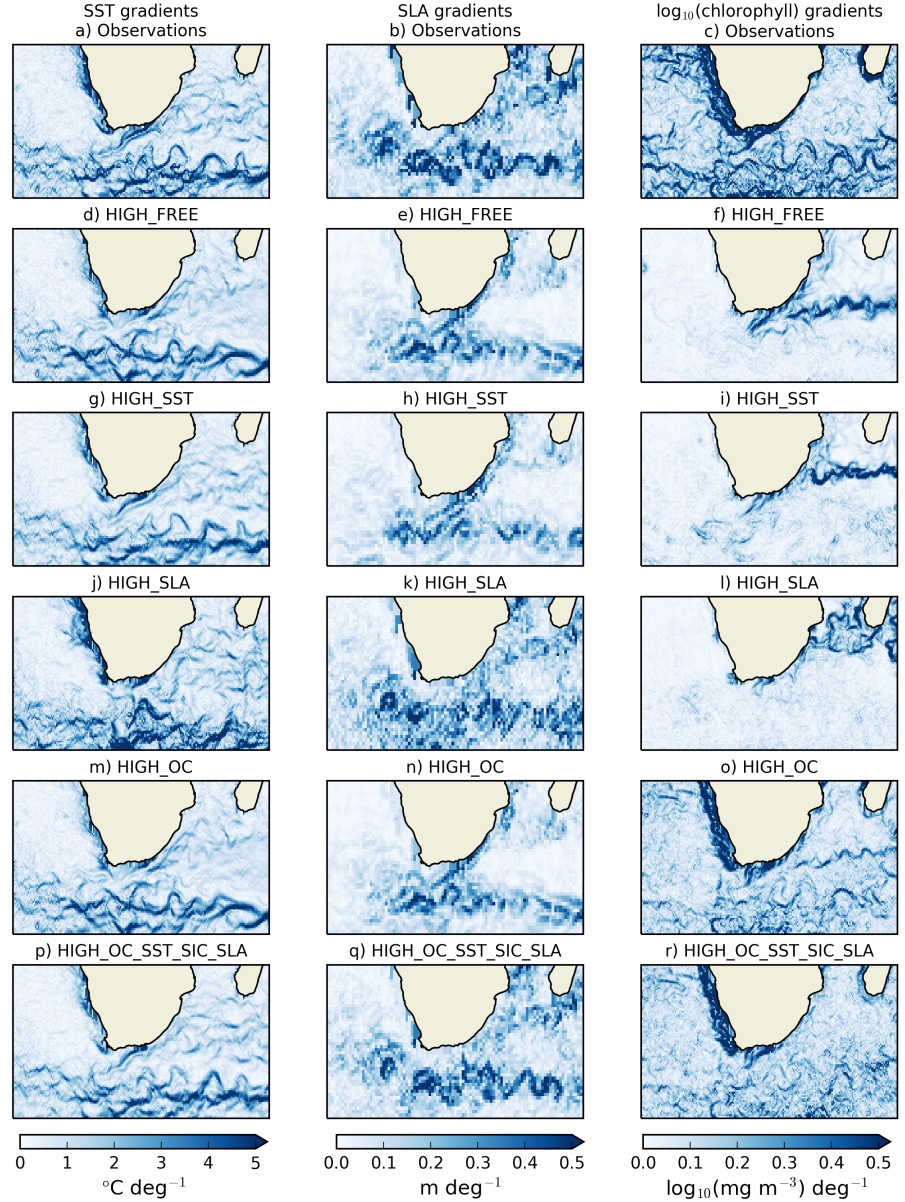

**Figure 4.** Observed and modelled gradients in the Agulhas Current region for December 2010.

as this gave clearest resolution of features for these variables, whilst SLA was binned onto a 1/2° grid, due to the lower

observational coverage. The Agulhas Current region was chosen due to its complex variability and physical-biogeochemical

interactions. Similar conclusions have been found from looking at other regions such as the Gulf Stream (not shown).

In the observation fields (Fig. 4a-c) a good correspondence can be seen between features in each variable. While the position

of gradients is not expected to be identical in all fields, SST fronts can generally be found around the eddies identified in the





SLA fields, with the $\log_{10}$(chlorophyll) gradients showing bloom activity along these fronts, relating to advection of nutrients
(Machu et al., 2005). This suggests that the CCI products are giving a suitably complementary view of such features.

In HIGH_FREE (Fig. 4d-f), SST and SLA gradients are found in similar locations to the observations, but not quite as
sharp, and the ocean is less energetic, as expected from an eddy-permitting resolution model. High primary production as-
sociated with the Agulhas retroflection is captured well, but otherwise the $\log_{10}$(chlorophyll) gradients are a poor match for
the observed fields. Gradients associated with SST fronts are too weak, and Southern Ocean primary production is too high
and overly homogeneous, leading to a spuriously strong gradient at the boundary of the Indian Ocean and Southern Ocean. In
HIGH_SST (Fig. 4g-i) improvements are seen in the positions of both SST and SLA gradients. Some small strengthening of
$\log_{10}$(chlorophyll) gradients is seen. In HIGH_SLA (Fig. 4j-l) the ocean becomes more energetic and the match with observed
SLA gradients is improved, but the impact on SST and $\log_{10}$(chlorophyll) gradients is mixed. In HIGH_OC (Fig. 4m-o) there is
no impact on physical fields due to the one-way coupling in the model, but $\log_{10}$(chlorophyll) gradients are greatly improved,
although still weaker than in the observations. In HIGH_OC_SST_SIC_SLA (Fig. 4p-r), the best match for observed gradients
of both SST and SLA can be seen. The $\log_{10}$(chlorophyll) gradients are similar in magnitude but overly noisy compared with
HIGH_OC, likely due to excessive vertical mixing resulting from the assimilation of SLA data, but the spurious gradient in
production between the Southern Ocean and Indian Ocean is finally removed.

## 5.3 Sea ice extent (Fig. 5)

Consistency is expected between SST and SIC in satellite products and analyses derived from them (Roberts-Jones et al.,
2012). Consistency should also be expected between OC and SIC products. Simply in terms of observational coverage, OC
cannot be observed under ice cover, and nearby ice may affect satellite retrievals (Bélanger et al., 2007), so OC data should
not be expected at locations where the OC retrieval would be contaminated by sea ice (Wang and Shi, 2009). Furthermore,
phytoplankton blooms often occur around the ice edge, where freshly melted ice reveals nutrient-rich stratified waters (Perrette
et al., 2011). These would be expected to be identified in observed and modelled chlorophyll fields, in locations commensurate
with the ice edge in SIC products.

Maps of SIC and surface chlorophyll in the Arctic are plotted from the observation products and a selection of 1/4° model
runs for the eight-day period from 17–24 May 2010 (Fig. 5). This period was chosen as it is during the spring bloom and ice
melt in the Arctic, has good observational coverage, and contains interesting and representative features. Daily ocean colour
products have insufficient spatial coverage, but the eight-day composite product for this period has near-complete coverage
while still representing a snapshot of conditions.

In the observation products (Fig. 5a-b), the limit of OC coverage mostly matches the ice edge, as defined by the 0.15 contour
(Parkinson and Cavalieri, 2008) and overlaid in red on the plots. In a few locations there are OC observations collocated with
sea ice, but only where the SIC is low. Furthermore, the eight-day OC product is a composite, while the mean of the L4 SIC
products over the period has been plotted, so this may be due to variability in SIC over the eight days. Overall, OC coverage
can be considered consistent with the SIC fields. Chlorophyll blooms can also be seen near to the ice edge in the Barents Sea
and Bering Sea.



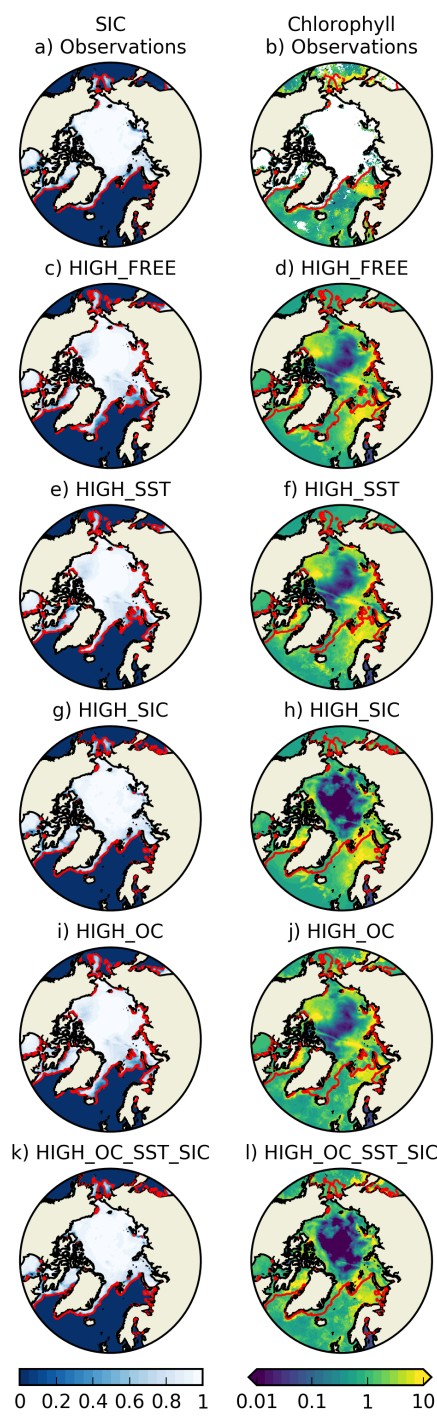

**Figure 5.** SIC (left column) and surface chlorophyll (right column) for 17–24 May 2010, from observed (a-b) and modelled (c-l) fields.





In HIGH_FREE (Fig. 5c-d), SIC is a reasonable qualitative match for the observations, but the concentrations are generally too low, and the ice edge too far south. There is a chlorophyll bloom in the Barents Sea, near the ice edge in the model,

but not in the Bering Sea. Within the ice-covered region, high chlorophyll concentrations are found in areas with moderate SIC. Observational studies have confirmed the presence of chlorophyll blooms under sea ice (Arrigo et al., 2012), but no observations are available for model validation.

In HIGH_SST, the position of the ice edge is a much better match for the observations than in HIGH_FREE. This suggests that the CCI SST products are consistent with the SIC products, and that assimilating SST therefore improves SIC. Chlorophyll

fields remain similar, except that the bloom in the Barents Sea extends further north, up to the altered ice edge.

HIGH_SIC has a further improved ice edge position, and higher SIC within the ice region, better matching the observations. This increase in SIC has the effect of reducing chlorophyll concentration in these areas. The lack of in situ observations means this change cannot be validated, but lower chlorophyll associated with greater ice cover is a consistent response due to the decrease in light availability.

HIGH_OC improves chlorophyll near to and away from the ice edge, including introducing bloom activity in the Bering Sea. Since SIC is unconstrained though, the assimilative model cannot capture exact details around the ice edge. Nor is any change seen within the ice region, where there are no OC observations.

In HIGH_OC_SST_SIC, the best match for the SIC and OC observations is seen, as well as reduced bloom activity in the ice region.

## 5.4 Carbon cycle validation (Fig. 6)

A key aim of global marine biogeochemical reanalysis is to study the carbon cycle, and provide the best possible estimates of surface $fCO_2$ and air-sea $CO_2$ flux. This can inform study of the global carbon budget (e.g., Le Quéré et al., 2018).

To assess the impact of assimilating different CCI products on the model's ability to reproduce surface $fCO_2$, each run has been validated against observations from the SOCAT v2 database (Bakker et al., 2014). The observations were passed to the

model while it was running, and an observation operator used which bilinearly interpolated the model fields to the observation locations at the nearest model time step to the observation time. From these match-ups, validation statistics have been calculated and displayed as a Taylor plot (Taylor, 2001) in Fig. 6. As in Ford and Barciela (2017), observations in shelf seas, defined as waters where the bottom depth is < 200 m (Simpson and Sharples, 2012), have been excluded from the validation. This is because a relatively coarse global model without tides is unable to represent complex shelf sea processes. In Fig. 6 a point is

plotted for each model run in Table 1, comparing to all off-shelf $fCO_2$ observations during the length of that run (1998–2010 for the 1° runs, 2008–2010 for the 1/4° runs). Furthermore, a point has been plotted for each of the 1° runs just assessing the years 2008–2010, allowing a direct comparison between the 1° and 1/4° runs.

At each resolution, assimilating OC or SIC products resulted in a small improvement in $fCO_2$ statistics, while assimilating SST resulted in a small degradation. Assimilating all three together improved the normalised standard deviation while lowering

the correlation. Assimilating SLA products, either individually or in combination with other variables, resulted in a large degradation in $fCO_2$ statistics. This is due to the impact on vertical mixing processes mentioned in the introduction.





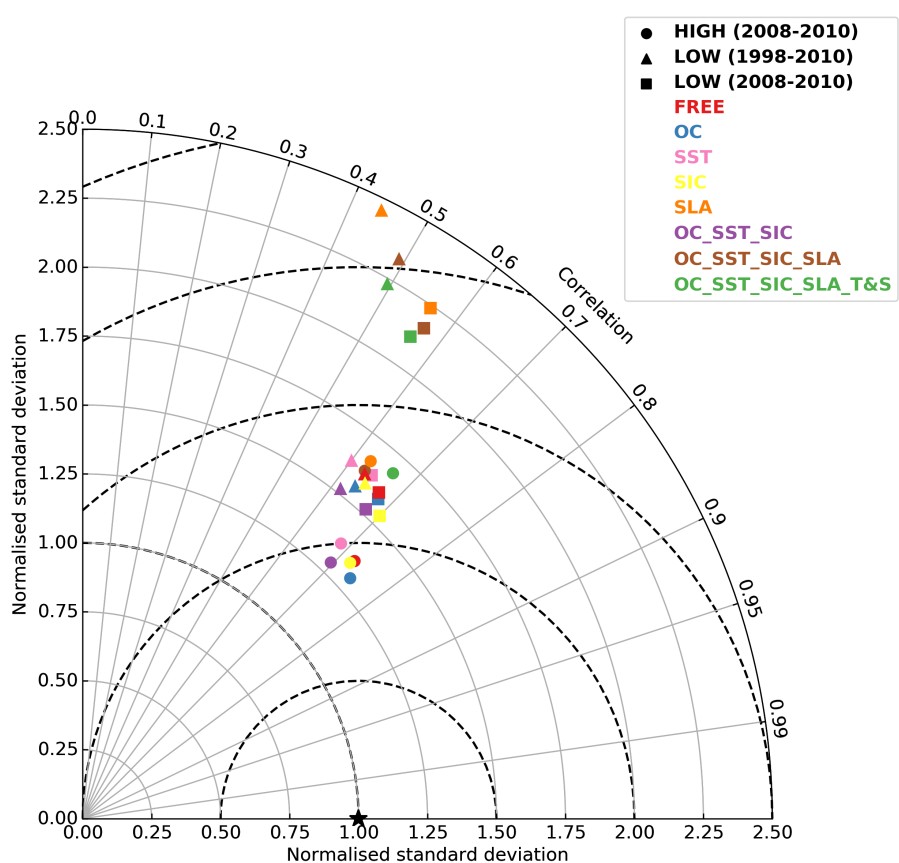

**Figure 6.** Taylor plot showing global fCO$_2$ statistics against SOCAT v2 for each run.

Comparing the 1° and 1/4° runs for 2008–2010, for all combinations of assimilation the 1/4° runs show a small but marked improvement in fCO$_2$ statistics. Given that the double-penalty effect (Gilleland et al., 2009) often masks improvements due to model resolution, this is a clear suggestion that the higher resolution model is better able to accurately represent fCO$_2$.

## 5.5 Phytoplankton and air-sea heat flux (Fig. 7)

One of the most dramatic and important features of the marine ecosystem is the spring bloom, and interannual variability in this can have wide-ranging impacts from carbon storage to fish stocks. Debate continues as to the exact mechanism which causes the bloom to occur (Behrenfeld and Boss, 2014, 2018), but some studies have suggested a direct link between the timing of the annual increase in phytoplankton and the timing of the net air-sea heat fluxes switching from negative to positive (Taylor and





Ferrari, 2011). Brody et al. (2013) compared different types of phenological metrics to determine the bloom start date in the
       North Atlantic from OC chlorophyll data, and whether this coincided with the switch to net heat uptake by the ocean. In the
       subtropics, where phytoplankton growth is not generally light-limited, they found the bloom consistently began well before the
       switch to net heat uptake. In the subpolar regions, the annual increase in chlorophyll concentration typically began at a similar
       time to the switch to net heat uptake, but with spatial variability across the region. Smyth et al. (2014) studied a long time series
of in situ data in the English Channel, and found that phytoplankton abundance consistently began to increase immediately
       following the switch to net heat uptake, explaining the interannual variability in bloom timings. At the North Atlantic Bloom
       Experiment site meanwhile, Mahadevan et al. (2012) found chlorophyll started to increase before the switch to net heat uptake.
       Contrasting conclusions may in part be due to some studies looking at chlorophyll concentration, and others at phytoplankton
       biomass (Westberry et al., 2016; Behrenfeld and Boss, 2018). The relationship between phytoplankton and net air-sea heat flux
at other stages of the seasonal cycle also remains an open question.

       To explore this relationship over a long model time series and throughout the seasonal cycle, and the impact on this of
       assimilating SST and OC data, maps of the correlation between surface chlorophyll concentration and net air-sea heat flux,
       and surface phytoplankton biomass and net air-sea heat flux, are plotted in Fig. 7 for LOW_FREE, LOW_OC, LOW_SST, and
       LOW_OC_SST_SIC, calculated from the daily model outputs for 1998–2010.

In LOW_FREE (Fig. 7a-b), there is a moderate positive correlation between chlorophyll and net air-sea heat flux in the
       subpolar North Atlantic and North Pacific, the Southern Ocean, and patches in the tropics. Elsewhere, there is generally a
       small to moderate negative correlation. Between phytoplankton biomass and net air-sea heat flux, there is a moderate to strong
       positive correlation almost everywhere, except for low correlation in the subtropical gyres and Indian Ocean.

       In LOW_OC (Fig. 7c-d), the patterns of correlation between chlorophyll and net air-sea heat flux become more coherent
at low latitudes, with positive correlation in the eastern Tropical Pacific and central Atlantic, and negative correlation in the
       subtropical gyres, western Tropical Pacific, and Indian Ocean. These patterns resemble patterns of mean chlorophyll concentra-
       tion (not shown). The correlation between phytoplankton biomass and net air-sea heat flux is weakened globally, with negative
       correlations in the subtropical gyres, becoming more like the correlation between chlorophyll and net air-sea heat flux.

       In LOW_SST (Fig. 7e-f), both the chlorophyll and phytoplankton biomass correlations with net air-sea heat flux remain
largely unaltered from LOW_FREE, though slightly elevated in equatorial regions. Correlations in LOW_OC_SST_SIC (Fig.
       7g-h) largely resemble those in LOW_OC, with some small regional differences.

       The different patterns seen in LOW_FREE for chlorophyll and phytoplankton biomass suggest that seasonal variations in
       carbon-to-chlorophyll ratio play an important role, and that studies based solely on chlorophyll data may not provide a full
       understanding of the underlying processes. This would imply that models, which unlike observations can provide a year-
round gap-free representation of relevant variables, should be able to provide a valuable contribution to such studies, with data
       assimilation able to constrain them to match available observations. However, assimilating OC data weakened the correlation
       between phytoplankton biomass and net air-sea heat flux in the model, such that it became more like the correlation between
       chlorophyll and net air-sea heat flux. This may have been a realistic response, or it may have been an artefact of the data
       assimilation method. In the scheme used, when chlorophyll derived from OC is assimilated, the phytoplankton biomass is





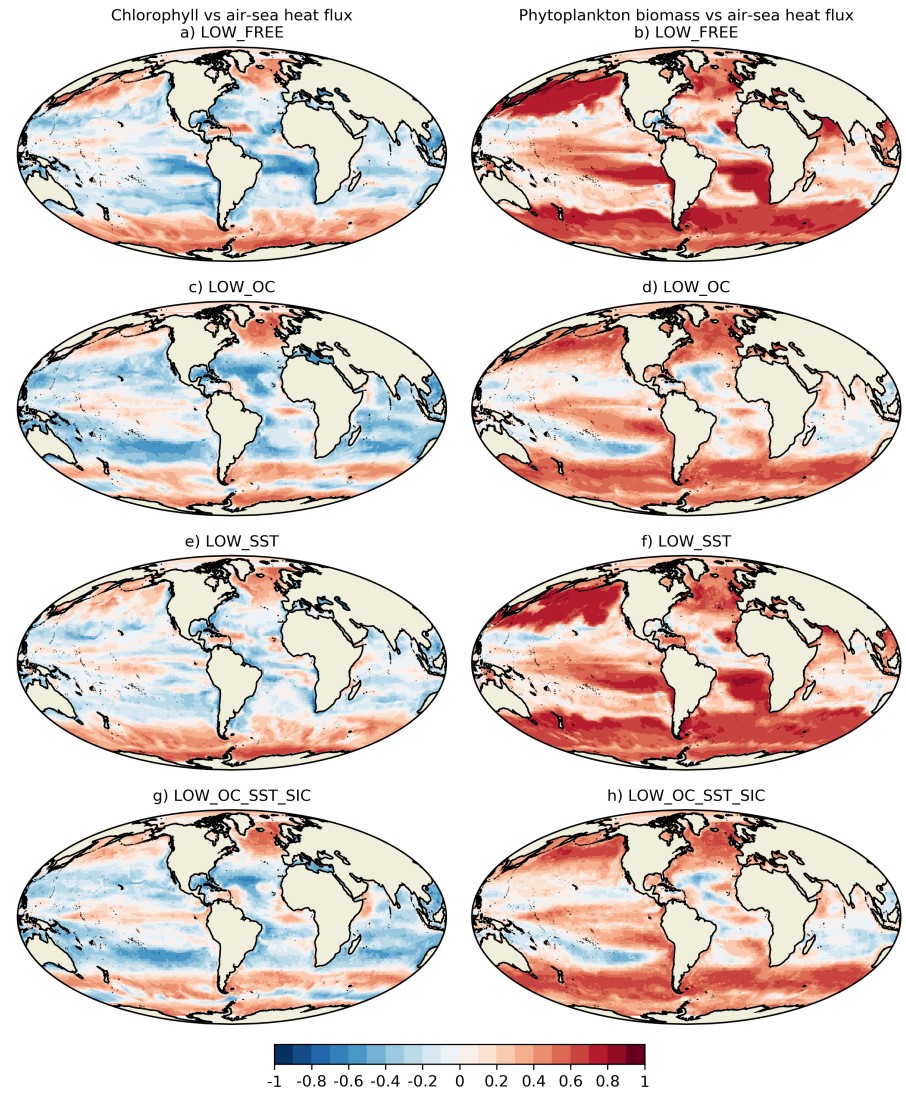

**Figure 7.** Maps of correlation between surface chlorophyll and net air-sea heat flux (left column) and surface phytoplankton biomass and net air-sea heat flux (right column), covering 1998–2010 for a selection of runs.

updated so as to maintain the existing carbon-to-chlorophyll ratio in the model. In essence, the assimilation assumes the carbon-to-chlorophyll ratio to be correct, and the phytoplankton biomass to be in error. However, it could be the carbon-to-chlorophyll ratio that is in error, or likely a mixture of both. Without widespread observations of phytoplankton biomass, this is difficult to assess, and therefore to know how to interpret the results. But if, for instance, a clear relationship could be determined between phytoplankton biomass and net air-sea heat flux, or another property, then the assimilation could be programmed to update the

carbon-to-chlorophyll ratio so as to maintain or enhance this relationship in the model.





## 6   Summary

A series of experiments was performed to assess the multivariate consistency, from a data assimilation perspective, of ESA CCI satellite observation products for ocean colour (OC), sea surface temperature (SST), sea level anomaly (SLA), and sea ice concentration (SIC). The products were assimilated, individually and in combination, into a global physical-biogeochemical ocean model, to create a set of 13-year reanalyses at 1° resolution and 3-year reanalyses at 1/4° resolution. These have been assessed through a series of case studies, examining the assimilation increments, frontal features, sea ice extent and phytoplankton blooms, carbon cycle validation, and the relationship between phytoplankton variability and air-sea heat fluxes.

In the assessment performed, the observation products were found to be consistent in terms of their representation of spatial features, such as fronts and eddies in the Agulhas Current region, and SST, OC and SIC around the retreating ice edge in the Arctic. This gives confidence that the ESA CCI products can be used in synergy to gain insights into the Earth system. In these cases, this consistency transferred through the data assimilation, resulting in an improved reanalysis when the satellite products were assimilated in combination, rather than just individually.

In other cases, assimilating particular variables served to degrade, rather than improve, the representation of other non-assimilated variables in the reanalysis. This was found to be due to issues with the model and data assimilation system, rather than any lack of consistency between the observations being assimilated. Much of this was down to known issues with physical data assimilation causing spurious vertical mixing, which is not unique to the system used in this study (While et al., 2010; Raghukumar et al., 2015; Park et al., 2018). But it also revealed complex interactions between the model and assimilation, with the assimilation of individual variables improving some non-assimilated variables while degrading others, and correcting some compensating errors while introducing others. This could potentially be counteracted by assimilating additional variables such as nutrients (Yu et al., 2018), but these cannot be observed from space and so observational coverage remains insufficient. The experiments performed in this study highlight ways in which the model and assimilation could each be developed accordingly, but there does not appear to be any one simple fix. These conclusions apply to both the 1° and 1/4° configurations of the model, though the higher resolution model was better able to simulate surface $fCO_2$, with and without data assimilation.

Previous studies have suggested a direct correlation between the timing of the initiation of the spring bloom and that of the annual switch from negative to positive air-sea heat fluxes (Taylor and Ferrari, 2011; Smyth et al., 2014). Other studies have reached contrasting (Mahadevan et al., 2012) or mixed (Brody et al., 2013) conclusions. This may in part be due to some studies looking at chlorophyll concentration, and others at phytoplankton biomass (Westberry et al., 2016; Behrenfeld and Boss, 2018). The reanalyses produced in this study provided an opportunity to look at this relationship in a long model time series, and the impact of data assimilation. In the free-running model, there was a strong positive correlation between phytoplankton biomass and net air-sea heat flux across much of the ocean, whereas for chlorophyll concentration the correlation with net air-sea heat flux was weaker, and often negative at low latitudes. This suggests that seasonal variations in carbon-to-chlorophyll ratio play an important role, and that studies of phytoplankton bloom initiation based solely on chlorophyll data may not provide a full understanding of the underlying processes. However, the assimilation of OC data weakened the correlation between phytoplankton biomass and net air-sea heat flux. In the absence of widespread in situ observations of phytoplankton biomass,



it is difficult to assess if this response was realistic, or due to a deficiency in the assimilation methodology. This serves to emphasise the importance of studying multivariate relationships, and considering both observational and model data.

*Data availability.* The nature of the 4D data generated in running the model experiments requires a large tape storage facility. These data are in excess of 100 terabytes (TB). However, the data can be made available upon request from the author.

*Author contributions.* DF designed and performed the experiments, analysed the results, and wrote the manuscript.

*Competing interests.* The author declares that they have no conflict of interest.

*Acknowledgements.* This study was funded by the European Space Agency (ESA) through the Climate Modelling User Group (CMUG) component of the Climate Change Initiative (CCI) project. Thank you to Jenny Machinford, Susan Kay, and Matt Martin for useful discussions and comments on the draft manuscript. Surface Ocean $CO_2$ Atlas (SOCAT) data were obtained from https://www.socat.info. SOCAT is an international effort, endorsed by the International Ocean Carbon Coordination Project (IOCCP), the Surface Ocean Lower Atmosphere
Study (SOLAS) and the Integrated Marine Biosphere Research (IMBeR) program, to deliver a uniformly quality-controlled surface ocean $CO_2$ database. The many researchers and funding agencies responsible for the collection of data and quality control are thanked for their contributions to SOCAT.



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
