# Peer review of "Assessing the role and consistency of satellite observation products in global physical-biogeochemical ocean reanalysis"

_Ocean Science, 2019_

## Referee Comment (RC1) · Anonymous Referee #1 · 18 Feb 2020

**General comments:**

The manuscript aims to investigate the impact of several new satellite products on global physical-biogeochemcial ocean reanalysis by a series of assimilation experiments. The work carried out analysis and comparisons in two period runs (13 and 3 years) and two model horizontal resolutions ( 1° and 1/4°). One of main conclusions is the satellite products and the reanalysis assimilating them are consistent in their representation of spatial features. Author also study the correlation relationship of net air-sea heat fluxes, phytoplankton biomass and chlorophyll concentration. Investigating the performance of new satellite products in a reanalysis is help for both numerical model development and ocean state analysis. Therefore, the topic in this study is relevant to the scope of Ocean Science. However, I think that the main points they should address are the following:

**Specific comments:**

1) Author did reanalysis with two model configurations and the same observations. However, there is nothing to compare these two runs in Section 5. Observations assimilated into different model configurations can resolve the observation representation of observations in explain some processes. Furthermore, the inter-comparison is help to study the consistency of observation and assimilation system in different horizontal resolution.

2) The manuscript analysed and compared the muti-year average of reanalysis results. This method for the analysis of the results is help to give the conclusion of the spatial features. However, it is also worth investigating the temporal features of these satellite products. Therefore, it is recommenced to address the study of temporal consistency of these products in the reanalysis and representation of major physical or biogeochemical process.

3) Author assessed the results with a series of cases studies. For example, in the section 5.2, the study only give one example in Agulhas. I don't think one example is enough to support the conclusion of fronts and eddies in the spatial consistency between satellite products and reanalysis. Numerical simulation may have performance in the different regions, especially for global numerical model.

4) This study assimilated several satellite products and in-situ observations. Most of the conclusions come from adding observation product to reanalysis individually or in combination. However, it lacks validation and analysis from independent observations except Section 5.4. this study is more like to focus on the impact of the observations from both satellite and in-situ on FOAM reanalysis system.

5) At the end of the spin up, you uniformly adjusted SSH to zero global mean for removing the SSH drift. I am concerned about some aspects of this method in simulation stability. I am curious whether the problem in your description at Line 161 is from this initial SSH setup.

6) In Section 2, the setup of model and data assimilation is too abbreviated, even lot of researches have been addressed based on FOAM system. Some important features and configurations need be detailed for easily reading.

**Minor comments:**

In abstract, author "Assimilating multiple variables together often resulted in larger mean increments for a variable than assimilating it individually, revealing ways in which the model and assimilation scheme could be improved." isn't consistent with that in summary. Further, it seems

"assimilating multiple variables …..." doesn't support "revealing ways in which the model and assimilation scheme could be improved."

L39-40 "It is not yet routine though to combine the assimilation of physics and biogeochemistry in a single ocean reanalysis." is not true. There are lots of reanlaysis products right now. For example, the CMEMS products….

Remove the "Fig--" in subsection title.

L 108-110, in-situ SST is mentioned. However, the specified influence of adding these observations hasn't been clarified in results.

L160-170 it needs to be detailed in the description of No data assimilation in some regions. For example, "the no increments were applied in the Malvinas Current region on a SMALL NUMBER of Dates", "No assimilation was performed on 18 January 2000 in 1° runs including SLA assimilation, as a few large SLA observations were causing the model to fail." Why assimilation is failed? If it is done by data assimilation system it will needs your further tune the data assimilation system before the reanalysis.

L 180 "The larger the increments, the larger the corrections  being applied to the model to keep it close to the observations."  In some cases , larger increment can cause model failed and leave far way the observations.

L 222-225 these description is contrary to the sentences L180-L182

Figure 4 needs to be improved and adds the coordinates

L247-248 "….SLA gradients is improved, but the impact on SST and log 10 (chlorophyll) gradients is mixed"

In Section 5.3. please show where both Barents Sea and Bering Sea are in plots

L294 Adding the description of the observations from SOCAT v2 database in Section 3

L315-325, the discussion don't need detailed and it is repeated in L 379-L382.

---

## Referee Comment (RC2) · Anonymous Referee #2 · 18 Apr 2020

First, I apologize for being so late with the review, due to exceptional circumstances.

The paper aims at assessing the impact of 4 remotely-sensed datasets on different versions of a global model, and in particular examine the consistency in between the datasets. The impact of assimilating one or multiple datasets into the models is examined in terms of different model variables. The datasets covers physics and biology, and the impact of physics on biology is also examined. The inverse is not examined, as there is no feedback from biology to physics. Some interesting -and sometimes maybe counter-intuitive- conclusions are obtained, e.g. that the assimilation of certain datasets degrades certain model variables.

[Figure]

The methods and the results are clearly described. Given that the models (physics and biogeochemical) and data assimilation methods are already extensively described and validated in previous papers, they form a very sound basis for the present study. There are some limitations or deficiencies in the modeling system, but they are known and acknowledged. Thus, it does not require to be validated again in the present paper. Some aspects of the data assimilation procedure feel a little like a cooking recipe, but I guess all modelling systems have these kind of safety nets in their implementation of the data assimilation (e.g. skip assimilation some days when it renders the model unstable, or at least play with the error covariances...).

In general, the paper is clearly structured, very well written, and I did not find typos. The paper is very interesting and timely, because it provides the kind of information needed for near-future versions of biogeochemical model simulations, both reanalysis and operational. As the author mentions, the impact of data assimilation of physics on biogeochemistry is still insufficiently studied, in particular methods for mitigating the impact of spurious vertical currents, but this point is not the topic of the present paper. This study about the remaining problems linked to co-assimilation of different variables is very welcome.

Minor remarks:

* although the author refers extensively to the relevant papers for the physical model, biogeochemical model, and data assimilation method, it could help to very briefly provide a few key facts. For example, for the physical model, he could mention the Nemo version (around line 62 page 3). For the BGC model, he could mention how many state variables there are, etc.

* the results seem valid for both the 1 degree and 1/4 degree models (independently from the fact that the higher resolution seems to present better results, even though the double penalty). But little detail is given regarding the resolution, and potentially, generalisations to even higher resolutions such as used in other regional models.

* can the author give precisions how his conclusions may ultimately lead to refinements or improvements in the models (as he says in the abstract that this is a potential benefit of the study) ? Is he thinking of methods like parameter estimations, or is he simply pointing to the known limitations of the current modelling system (such as biases mentionned at line 179, 189, 206).

* maybe the text about the safety-nets of the assimilation procedure can be re-written to make it feel more rigourous (as it certainly is)

---

## Author Comment (AC1) · 28 May 2020

**The manuscript aims to investigate the impact of several new satellite products on global physical-biogeochemcial ocean reanalysis by a series of assimilation experiments. The work carried out analysis and comparisons in two period runs (13 and 3 years) and two model horizontal resolutions ( 1° and 1/4°). One of main conclusions is the satellite products and the reanalysis assimilating them are consistent in their representation of spatial features. Author also study the correlation relationship of net air-sea heat fluxes, phytoplankton biomass and chlorophyll concentration. Investigating the performance of new satellite products in a reanalysis is help for both numerical model development and ocean state analysis. Therefore, the topic in this study is relevant to the scope of Ocean Science.**

Thank you for reviewing the manuscript, and for the supportive comments and constructive suggestions. I will respond to your comments in turn below.

**However, I think that the main points they should address are the following:**
**Specific comments:**
**1) Author did reanalysis with two model configurations and the same observations. However, there is nothing to compare these two runs in Section 5. Observations assimilated into different model configurations can resolve the observation representation of observations in explain some processes. Furthermore, the inter-comparison is help to study the consistency of observation and assimilation system in different horizontal resolution.**

In the original manuscript, comparison is made between the two resolutions in Section 5.4, which examines the representation of the carbon cycle. I propose adding a further comparison examining variability in the Tropical Pacific in response to ENSO in runs at each resolution. This extra comparison also addresses your next comment, so my proposed changes are detailed in my response to that.

I also propose to expand the discussion of resolution in the Summary and Conclusions section. In the original manuscript I simply stated:

*"These conclusions apply to both the 1° and 1/4° configurations of the model, though the higher resolution model was better able to simulate surface $fCO_2$, with and without data assimilation."*

I propose expanding this to:

*"Conclusions about model and assimilation performance, and consistency and variability, apply similarly to both the 1° and 1/4° configurations of the model. The higher resolution model was better able to simulate surface $fCO_2$, with and without data assimilation. This may be due to improved representation of processes in the 1/4° configuration, or may reflect differences in initialisation of DIC and alkalinity fields, which model $fCO_2$ has been shown to be sensitive to (Lebehot et al., 2019). The two resolutions show comparable temporal variability, with data assimilation having a similar impact. It is likely that conclusions about multivariate consistency are broadly generalisable to other resolutions and potentially regional models, though as all models and configurations have their own particular properties and biases, exact results may vary."*

*Lebehot, A. D., Halloran, P. R., Watson, A. J., McNeall, D. J., Ford, D. A., Landschützer, P., et al. (2019). Reconciling observation and model trends in North Atlantic surface $CO_2$. Global Biogeochemical Cycles, 33, 1204–1222. https://doi.org/10.1029/2019GB006186.*

**2) The manuscript analysed and compared the muti-year average of reanalysis results. This method for the analysis of the results is help to give the conclusion of the spatial features. However, it is also worth investigating the temporal features of these satellite products. Therefore, it is recommended to address the study of temporal consistency of these products in the reanalysis and representation of major physical or biogeochemical process.**

I propose to add a new sub-section and figure investigating the response of physical and biogeochemical fields to the El Niño Southern Oscillation (ENSO) in the Tropical Pacific, the impact of SST and ocean colour assimilation on this, and differences between model resolutions:

*5.x Temporal variability*

*"A major driver of climate variability is the El Niño Southern Oscillation (ENSO). One measure of ENSO variability is the Niño 3.4 index (Fig. xa), calculated as the five-month running mean of SST anomalies in the Niño 3.4 region (5°N-5°S, 170°W-120°W) of the Tropical Pacific (Trenberth, 1997). To explore the representation of ENSO variability in SST, vertically integrated primary production (PP) and air-sea $CO_2$ flux, and the impact of SST and OC assimilation and model resolution, five-month running means of these variables averaged over the Niño 3.4 region are plotted in Fig. x. For LOW_FREE and HIGH_FREE the absolute values are plotted, and for LOW_SST, HIGH_SST, LOW_OC, HIGH_OC, LOW_OC_SST_SIC, and HIGH_OC_SST_SIC, anomalies from LOW_FREE and HIGH_FREE are plotted.*

*ENSO variability in SST is well reproduced in LOW_FREE (Fig. xb), aligning with ENSO events seen in the observed Niño 3.4 SST index (Fig. xa) as calculated from HadISST1 data (Rayner et al., 2003) and downloaded from [https://psl.noaa.gov/gcos_wgsp/Timeseries/Data/nino34.long.anom.data](https://psl.noaa.gov/gcos_wgsp/Timeseries/Data/nino34.long.anom.data). HIGH_FREE shows very similar variability, but with slightly higher SST than LOW_FREE. In the first few years of LOW_SST, the assimilation acted to reduce SST compared to LOW_FREE (Fig. xc), enhancing the prolonged La Niña (negative Niño 3.4 SST index) conditions of the period. The assimilation also served to enhance the El Niño (positive Niño 3.4 SST index) of 2009/10, but otherwise largely just modulated seasonal variability of SST rather than interannual variability. In HIGH_SST there was a similar impact on variability, but the anomaly from HIGH_FREE is offset in magnitude from that between LOW_SST and LOW_FREE, with the assimilation consistently cooling the model.*

*Very low PP is seen in LOW_FREE (Fig. xd) at the beginning of the time series, related to the major El Niño event of 1997/98. Much more limited interannual variability is seen through the rest of the period, but with slightly reduced PP during the 2002/03 and 2009/10 El Niño events. Variability in HIGH_FREE is very similar to that in LOW_FREE, but slightly offset in magnitude, as with SST. Assimilating SST individually had limited impact on PP (Fig. xe), while assimilating OC individually served to substantially reduce PP and impact seasonal variability. In LOW_OC_SST_SIC the SST assimilation made more difference than in LOW_SST, including changing interannual variability during the 1998-2001 La Niña conditions. The difference between LOW_OC_SST_SIC and HIGH_OC_SST_SIC is frequently greater than the combined difference between LOW_SST and HIGH_SST, and between LOW_OC and HIGH_OC.*

*In air-sea $CO_2$ flux a clear ENSO signal is seen in LOW_FREE (Fig. xf), similar to that in SST. HIGH_FREE displays the same variability, but with a clear offset. The smaller offsets in SST and PP may contribute to this, but it is most likely caused by differences in the initialisation of DIC and alkalinity (Lebehot et al., 2019). Assimilation of OC data had little impact (Fig. xg), while assimilation of SST had an impact on the seasonal cycle, and slightly*

*reduced air-sea $CO_2$ flux anomalies during La Niña conditions. SST assimilation also served to increase the differences between LOW_FREE and HIGH_FREE."*

[Figure]

*Figure x: Five-month running mean time series of variables averaged over the Niño 3.4 region (5°N-5°S, 170°W-120°W). (a) Observed Niño 3.4 SST index (Trenberth, 1997) as calculated from HadISST1 data (Rayner et al., 2003) and downloaded from* https://psl.noaa.gov/gcos_wgsp/Timeseries/Data/nino34.long.anom.data. *(b) SST in free runs, (c) anomaly of SST from free runs, (d) vertically integrated primary production in free runs, (e) anomaly of vertically integrated primary production from free runs, (f) air-sea $CO_2$ flux in free runs, (g) anomaly of air-sea $CO_2$ flux from free runs.*

*Lebehot, A. D., Halloran, P. R., Watson, A. J., McNeall, D. J., Ford, D. A., Landschützer, P., et al. (2019). Reconciling observation and model trends in North Atlantic surface $CO_2$. Global Biogeochemical Cycles, 33, 1204–1222. https://doi.org/10.1029/2019GB006186.*

*Rayner N. A., D. E. Parker, E. B. Horton, C. K. Folland, L. V. Alexander, D. P. Rowell, E. C. Kent, A. Kaplan, Global analyses of sea surface temperature, sea ice, and night marine air temperature since the late nineteenth century, J. Geophys. Res., 108 (D14), 4407, doi:10.1029/2002JD002670, 2003.*

*Trenberth, K.E., 1997: The Definition of El Niño. Bull. Amer. Meteor. Soc., 78, 2771–2778, https://doi.org/10.1175/1520-0477(1997)078<2771:TDOENO>2.0.CO;2.*

**3) Author assessed the results with a series of cases studies. For example, in the section 5.2, the study only give one example in Agulhas. I don't think one example is enough to support the conclusion of fronts and eddies in the spatial consistency between satellite products and reanalysis. Numerical simulation may have performance in the different regions, especially for global numerical model.**

I can add a further example, expanding on the line in the original manuscript: *"Similar conclusions have been found from looking at other regions such as the Gulf Stream (not shown)."* I propose adding the following example:

*"In the Gulf Stream (Fig. 5), similar results were found. In the observation fields SST and $\log_{10}$(chlorophyll) fronts are largely collocated, and situated around eddies identified in the SLA products. In HIGH_FREE the SST gradients are broadly similar to the observed fields, but some specific features are lacking. SLA and $\log_{10}$(chlorophyll) gradients are found in corresponding locations, but too weak in magnitude compared to the observations. In HIGH_SST the position and magnitude of gradients is improved in all three fields. In HIGH_SLA the SLA gradients are improved, with some improvement to SST and $\log_{10}$(chlorophyll) gradients, but also increased noise. In HIGH_OC the location of $\log_{10}$(chlorophyll) gradients match those in the observed fields, but the magnitude remains too weak. In HIGH_OC_SST_SLA the best combined representation of gradients in the three fields is seen."*

[Figure]

*Figure 5. Observed and modelled gradients in the Gulf Stream region for December 2010.*

**4) This study assimilated several satellite products and in-situ observations. Most of the conclusions come from adding observation product to reanalysis individually or in combination. However, it lacks validation and analysis from independent observations except Section 5.4. this study is more like to focus on the impact of the observations from both satellite and in-situ on FOAM reanalysis system.**

The lack of validation was a deliberate choice, as this was outside the scope of the study, and all components of the system have been previously validated in the literature. As stated in the manuscript:

*"The model runs have been assessed through a series of case studies, presented in turn below. These are intended to explore physical-biogeochemical relationships in the model and observations, and the impact of data assimilation on these, rather than simply validating the accuracy of the reanalyses. For validation of the underlying system, the reader is referred to Blockley et al. (2014) for the physical model and assimilation, Ford and Barciela (2017) for the biogeochemical model and assimilation, and Lea et al. (2014) for data withholding experiments performed with the physics-only system."*

In relation to this, Referee #2 commented:

*"Given that the models (physics and biogeochemical) and data assimilation methods are already extensively described and validated in previous papers, they form a very sound basis for the present study. There are some limitations or deficiencies in the modeling system, but they are known and acknowledged. Thus, it does not require to be validated again in the present paper."*

I therefore propose not to add any extra validation with independent observations, beyond that already presented in Section 5.4, in line with the original aims of the study and the comments of Referee #2. However, I do propose to extend the text above to include a further relevant reference, to the recently published study of King et al. (2020):

*"For validation of the underlying system, the reader is referred to Blockley et al. (2014) for the physical model and assimilation, Ford and Barciela (2017) for the biogeochemical model and assimilation, and Lea et al. (2014) and King et al. (2020) for data withholding experiments performed with the physics-only system."*

*King, RR, Lea, DJ, Martin, MJ, Mirouze, I, Heming, J. The impact of Argo observations in a global weakly coupled ocean–atmosphere data assimilation and short-range prediction system. Q J R Meteorol Soc. 2020; 146: 401– 414. https://doi.org/10.1002/qj.3682.*

**5) At the end of the spin up, you uniformly adjusted SSH to zero global mean for removing the SSH drift. I am concerned about some aspects of this method in simulation stability. I am curious whether the problem in your description at Line 161 is from this initial SSH setup.**

This procedure had no impact on model dynamics, as the adjustment to SSH was uniform and all gradients and features remained identical. In a free run, the only impact seen on subsequent model results would be a constant offset in the SSH field, all other model variables would be identical. It only affects results in a run assimilating SLA data by removing a constant model bias which would have resulted in large biased increments at the start of the run. I propose to clarify this by changing the original text from:

*"At the end of the spin-up, the NEMO sea surface height (SSH) fields were uniformly adjusted to have zero global mean, as the global mean SSH had drifted and would have caused a large initialisation shock when SLA assimilation began."*

to:

*"At the end of the spin-up, a uniform constant was added to the NEMO sea surface height (SSH) fields to give a global mean SSH of zero, as the global mean SSH had drifted and would have caused a large initialisation shock when SLA assimilation began. This procedure maintained all SSH gradients and features, and had no impact on model dynamics."*

The problem described at Line 161 of the original manuscript was entirely unrelated, and due to a few anomalously large SLA observations in the version of sea level products used (these have been corrected in subsequent versions). I propose to clarify this by modifying the original text from:

*"No assimilation was performed on 18 January 2000 in 1° runs including SLA assimilation, as a few large SLA observations were causing the model to fail."*

to:

*"No assimilation was performed on 18 January 2000 in 1° runs including SLA assimilation, as a few anomalously large SLA observations resulted in unrealistic increments."*

**6) In Section 2, the setup of model and data assimilation is too abbreviated, even lot of researches have been addressed based on FOAM system. Some important features and configurations need be detailed for easily reading.**

I propose adding the following details to the text: the NEMO version and number of HadOCC state variables (as suggested by Referee #2), the one-way coupling between physics and biogeochemistry, the use of first guess at appropriate time (FGAT) by the data assimilation, and the use of conservation of mass and estimates of phytoplankton growth and loss errors by the nitrogen balancing scheme.

**Minor comments:**
**In abstract, author "Assimilating multiple variables together often resulted in larger mean increments for a variable than assimilating it individually, revealing ways in which the model and assimilation scheme could be improved." isn't consistent with that in summary. Further, it seems "assimilating multiple variables …..." doesn't support "revealing ways in which the model and assimilation scheme could be improved."**

I think my intended meaning has been misunderstood, so I propose to rephrase this sentence to try and clarify:

*"Assimilating multiple variables together often resulted in larger mean increments for a variable than assimilating it individually, providing information about model biases and compensating errors which could be addressed in future development of the model and assimilation scheme."*

**L39-40 "It is not yet routine though to combine the assimilation of physics and biogeochemistry in a single ocean reanalysis." is not true. There are lots of reanlaysis products right now. For example, the CMEMS products….**

I agree that there are plenty of reanalysis products, but most of these, including most of the CMEMS products, do not yet include assimilation of both physical and biogeochemical observations. To make my intended meaning clearer, I propose to rephrase the quoted sentence:

*"It is not yet routine though for a single ocean reanalysis to include the assimilation of both physical and biogeochemical data."*

**Remove the "Fig--" in subsection title.**

I will remove these as requested.

**L 108-110, in-situ SST is mentioned. However, the specified influence of adding these observations hasn't been clarified in results.**

As stated in Section 4, the influence of in situ SST observations has been considered in combination with the influence of in situ temperature and salinity profiles, so can be seen in results including the runs HIGH_OC_SST_SIC_SLA_T&S and LOW_OC_SST_SIC_SLA_T&S. In order to separate out the influence of different sources of in situ observations would require extra model runs to be performed. This is outside the scope of the present study, which is specifically focussed on satellite observations.

**L160-170 it needs to be detailed in the description of No data assimilation in some regions. For example, "the no increments were applied in the Malvinas Current region on a SMALL NUMBER of Dates", "No assimilation was performed on 18 January 2000 in 1° runs including SLA assimilation, as a few large SLA observations were causing the model to fail." Why assimilation is failed? If it is done by data assimilation system it will needs your further tune the data assimilation system before the reanalysis.**

The issue with SLA assimilation on 18 January 2000 is addressed in a previous comment above. I agree that this section could have been better written, and should be modified. The original text reads:

*"In most cases, assimilation increments were applied at all model grid points. However, for model stability a few exceptions were required. No increments were applied in the Baltic Sea in the 1° runs, which is treated as an enclosed sea at this resolution. No assimilation was performed on 18 January 2000 in 1° runs including SLA assimilation, as a few large SLA observations were causing the model to fail. On a few occasions the assimilation caused LOW_SLA and LOW_OC_SST_SIC_SLA_T&S to fail near the Antarctic coast; in these cases no increments were applied for a short period in the surrounding region. Similarly, no increments were applied in the Malvinas Current region on a small number of dates in HIGH_SLA and HIGH_OC_SST_SIC_SLA_T&S. On all dates, no biogeochemical increments were applied in grid boxes with SIC greater than 0.01, which is a relaxation of the conditions imposed by Ford et al. (2012) and Ford and Barciela (2017). Furthermore, phytoplankton nitrogen increments were limited in magnitude to 1.0 mmol m$^{-3}$ in a region surrounding the Amazon river outflow, prior to running the Hemmings et al. (2008) nitrogen balancing scheme, in order to avoid spuriously large DIC increments at very low chlorophyll concentrations. These cases were generally indicative of issues with the model and assimilation procedure under specific circumstances, rather than of errors in the observation products."*

I propose rewriting this as follows:

*"In most cases, assimilation increments were applied at all model grid points. However, for model stability the following exceptions were required:*
- *No increments were applied in the Baltic Sea in the 1° runs, as it is treated as an enclosed sea at this resolution.*

- *No biogeochemical increments were applied in grid boxes with SIC greater than 0.01, in a relaxation of the conditions imposed by Ford et al. (2012) and Ford and Barciela (2017).*
- *Phytoplankton nitrogen increments were limited in magnitude to 1.0 mmol $m^{-3}$ in a region surrounding the Amazon river outflow, prior to running the Hemmings et al. (2008) nitrogen balancing scheme. This was to avoid spuriously large DIC increments at very low chlorophyll concentrations in the region of freshwater influence.*
- *No assimilation was performed on 18 January 2000 in 1° runs including SLA assimilation, as a few anomalously large SLA observations resulted in unrealistic increments.*
- *Near the Antarctic coast during February and March, sparse SLA and T&S observations located in melt ponds occasionally led to unrealistically large increments being generated in LOW_SLA and LOW_OC_SST_SIC_SLA_T&S. In these cases, no increments were applied for a short period in the surrounding region until the ice had melted further.*
- *SLA assimilation is designed to be performed in combination with T&S assimilation (Lea et al., 2014), and assimilating SLA data on its own can sometimes result in adverse changes to subsurface density structure in energetic regions. This occasionally led to a model instability in the Malvinas Current region in HIGH_SLA, and so to prevent this no increments were applied in this region on 12 dates during the run. This was also required on one date during HIGH_OC_SST_SIC_SLA_T&S."*

**L 180 "The larger the increments, the larger the corrections being applied to the model to keep it close to the observations." In some cases , larger increment can cause model failed and leave far way the observations.**

This can, occasionally, be the unintended consequence of large assimilation increments in the case of a model blowing up. Unless there is a bug in the assimilation scheme though, the assimilation is still attempting to bring the model closer to the observations, and it is a model instability which leads to the opposite result. I therefore propose making a small change to the quoted sentence so that it reads:

*"The larger the increments, the larger the corrections being applied to the model to try to keep it close to the observations."*

**L 222-225 these description is contrary to the sentences L180-L182**

Lines 180-182 stated:

*"In theory, if the observation products are providing consistent information, and the model and assimilation scheme are performing as intended, then assimilating multiple ECVs should result in smaller mean increments for a given ECV compared with assimilating that ECV alone."*

Lines 222-225 stated:

*"Given these issues, looking at mean increments does not provide evidence either way about whether the CCI products are mutually consistent, but it does highlight issues with the multivariate assimilation which can be addressed during future development work. It should also be noted that the physics data assimilation is designed to work best when all data types are available, as these provide complementary information (Lea et al., 2014)."*

I would argue that these two sections are consistent, but accept that some rephrasing is required for this to be clearer to the reader. In the first section two conditions are stated: "*if the observation products are providing consistent information*" and "*the model and assimilation scheme are performing as intended*". The latter section was intended to convey that the second of these conditions was not satisfied, and therefore no conclusion could be drawn as to whether the first condition was satisfied. In order to clarify this, I propose rephrasing the latter section:

*"The finding that assimilating multiple ECVs often results in larger mean increments for a given ECV compared with assimilating that ECV alone implies that either the observation products are providing inconsistent information, or that the model and assimilation are not performing entirely as intended. Analysis suggests the latter to be the case, meaning that looking at mean increments does not provide evidence either way about whether the CCI products are mutually consistent. It does though highlight issues with the multivariate assimilation which can be addressed during future development work. It should also be noted that the physics data assimilation is designed to work best when all data types are available, as these provide complementary information (Lea et al., 2014)."*

**Figure 4 needs to be improved and adds the coordinates**

I can add coordinates to the plots:

[Figure]

**L247-248 "….SLA gradients is improved, but the impact on SST and log 10 (chlorophyll) gradients is mixed"**

I am not clear what is being suggested here.

**In Section 5.3. please show where both Barents Sea and Bering Sea are in plots**

I propose adding an annotation as in the below:

[Figure]

*Figure 5. SIC (left column) and surface chlorophyll (right column) for 17–24 May 2010, from observed (a-b) and modelled (c-l) fields. In (b) chlorophyll blooms in the Bering Sea and Barents Sea are marked with "Be" and "Ba" respectively.*

**L294 Adding the description of the observations from SOCAT v2 database in Section 3**

These observations are already described in Section 3 of the original manuscript, on Lines 113-115.

**L315-325, the discussion don't need detailed and it is repeated in L 379-L382.**

I propose to remove most of the text in Lines 315-325, but keep the citations. The opening paragraph of the sub-section would therefore be shortened to:

*"One of the most dramatic and important features of the marine ecosystem is the spring bloom, and interannual variability in this can have wide-ranging impacts from carbon storage to fish stocks. Debate continues as to the exact mechanism which causes the bloom to occur (Behrenfeld and Boss, 2014, 2018), but some studies have suggested a direct link between the timing of the annual increase in phytoplankton and the timing of the net air-sea heat fluxes switching from negative to positive (Taylor and Ferrari, 2011; Smyth et al. 2014). Other studies have reached contrasting (Mahadevan et al., 2012) or mixed (Brody et al., 2013) conclusions. This may in part be due to some studies looking at chlorophyll concentration, and others at phytoplankton biomass (Westberry et al., 2016; Behrenfeld and Boss, 2018). The relationship between phytoplankton and net air-sea heat flux at other stages of the seasonal cycle also remains an open question."*

---

## Author Comment (AC2) · 28 May 2020

Response to Referee #2

**First, I apologize for being so late with the review, due to exceptional circumstances.**

**The paper aims at assessing the impact of 4 remotely-sensed datasets on different versions of a global model, and in particular examine the consistency in between the datasets. The impact of assimilating one or multiple datasets into the models is examined in terms of different model variables. The datasets covers physics and biology, and the impact of physics on biology is also examined. The inverse is not examined, as there is no feedback from biology to physics. Some interesting -and sometimes maybe counter-intuitive- conclusions are obtained, e.g. that the assimilation of certain datasets degrades certain model variables.**

**The methods and the results are clearly described. Given that the models (physics and biogeochemical) and data assimilation methods are already extensively described and validated in previous papers, they form a very sound basis for the present study. There are some limitations or deficiencies in the modeling system, but they are known and acknowledged. Thus, it does not require to be validated again in the present paper. Some aspects of the data assimilation procedure feel a little like a cooking recipe, but I guess all modelling systems have these kind of safety nets in their implementation of the data assimilation (e.g. skip assimilation some days when it renders the model unstable, or at least play with the error covariances...).**

**In general, the paper is clearly structured, very well written, and I did not find typos. The paper is very interesting and timely, because it provides the kind of information needed for near-future versions of biogeochemical model simulations, both reanalysis and operational. As the author mentions, the impact of data assimilation of physics on biogeochemistry is still insufficiently studied, in particular methods for mitigating the impact of spurious vertical currents, but this point is not the topic of the present paper. This study about the remaining problems linked to co-assimilation of different variables is very welcome.**

Thank you for your review of the manuscript, and your positive comments.

**Minor remarks:**
**\* although the author refers extensively to the relevant papers for the physical model, biogeochemical model, and data assimilation method, it could help to very briefly provide a few key facts. For example, for the physical model, he could mention the Nemo version (around line 62 page 3). For the BGC model, he could mention how many state variables there are, etc.**

The NEMO version is 3.4, and HadOCC has six state variables. I propose expanding the description in Section 2 to include this information, as well as additional details regarding the one-way coupling between physics and biogeochemistry, the use of first guess at appropriate time (FGAT) by the data assimilation, and the use of conservation of mass and estimates of phytoplankton growth and loss errors by the nitrogen balancing scheme.

**\* the results seem valid for both the 1 degree and 1/4 degree models (independently from the fact that the higher resolution seems to present better results, even though the double penalty). But little detail is given regarding the resolution, and potentially, generalisations to even higher resolutions such as used in other regional models.**

I propose adding an extra sub-section of the Results section examining variability in the Tropical Pacific in response to ENSO in runs at each resolution. Please see my response to

Referee #1 for details. I also propose to expand the discussion of resolution in the Summary and Conclusions section. In the original manuscript I simply stated:

*"These conclusions apply to both the 1° and 1/4° configurations of the model, though the higher resolution model was better able to simulate surface $fCO_2$, with and without data assimilation."*

I propose expanding this to:

*"Conclusions about model and assimilation performance, and consistency and variability, apply similarly to both the 1° and 1/4° configurations of the model. The higher resolution model was better able to simulate surface $fCO_2$, with and without data assimilation. This may be due to improved representation of processes in the 1/4° configuration, or may reflect differences in initialisation of DIC and alkalinity fields, which model $fCO_2$ has been shown to be sensitive to (Lebehot et al., 2019). The two resolutions show comparable temporal variability, with data assimilation having a similar impact. It is likely that conclusions about multivariate consistency are broadly generalisable to other resolutions and potentially regional models, though as all models and configurations have their own particular properties and biases, exact results may vary."*

*Lebehot, A. D., Halloran, P. R., Watson, A. J., McNeall, D. J., Ford, D. A., Landschützer, P., et al. (2019). Reconciling observation and model trends in North Atlantic surface $CO_2$. Global Biogeochemical Cycles, 33, 1204–1222. https://doi.org/10.1029/2019GB006186.*

**\* can the author give precisions how his conclusions may ultimately lead to refinements or improvements in the models (as he says in the abstract that this is a potential benefit of the study) ? Is he thinking of methods like parameter estimations, or is he simply pointing to the known limitations of the current modelling system (such as biases mentionned at line 179, 189, 206).**

I was thinking of both these types of things, and agree this could be expanded on in the manuscript. The Summary section contains the following text:

[revised manuscript text omitted]